# Assessing Variable Importance for Best Subset Selection

**DOI:** 10.3390/e26090801

**Published:** 2024-09-19

**Authors:** Jacob Seedorff, Joseph E. Cavanaugh

**Affiliations:** Department of Biostatistics, College of Public Health, University of Iowa, 145 N. Riverside Dr., Iowa City, IA 52242, USA; joe-cavanaugh@uiowa.edu

**Keywords:** AIC, BIC, feature selection, parametric bootstrap, post-selection inference, variable selection

## Abstract

One of the primary issues that arises in statistical modeling pertains to the assessment of the relative importance of each variable in the model. A variety of techniques have been proposed to quantify variable importance for regression models. However, in the context of best subset selection, fewer satisfactory methods are available. With this motivation, we here develop a variable importance measure expressly for this setting. We investigate and illustrate the properties of this measure, introduce algorithms for the efficient computation of its values, and propose a procedure for calculating *p*-values based on its sampling distributions. We present multiple simulation studies to examine the properties of the proposed methods, along with an application to demonstrate their practical utility.

## 1. Introduction

The assessment of the relative importance of each variable in a model is a problem of fundamental relevance in statistical modeling. Multiple approaches have been proposed for this purpose (see [1,2,3], and the references contained therein). Popular methods include scaling all variables to have a common standard deviation and then using coefficient estimates from the full model to assess importance. Another familiar technique is to use *p*-values for the coefficients from the full model to quantify importance. However, neither of these techniques are defensible when determining variable importance in the context of best subset selection. In fact, the relative importance of the variables in the full model may differ substantially from the importance of the corresponding variables in a best subset selection framework. For example, a variable may be statistically significant at the 0.05 level in the full model, but may also be excluded from the selected model.

Using the coefficient estimates for standardized variables or their associated *p*-values to evaluate variable importance for the selected model is equally problematic. In particular, for the variables that remain in the selected model, it is well known that the standard errors of coefficient estimates will be unrealistically small, leading to confidence intervals that are too narrow, inflated test statistics, and deflated *p*-values [4].

When performing best subset selection, the assessment of variable importance is more challenging. Ideally, any such method would quantify importance based on the variables that tend to be prevalent in the models favored by the criteria used to score the models. A cogent and intuitively appealing method proposed by Efron [4] uses the nonparametric bootstrap method to repeatedly perform best subset selection on bootstrapped datasets and quantifies importance based on the proportion of times each variable is included in the optimal model. However, this approach entails performing best subset selection many times over, which will be computationally expensive even with the advent of branch and bound algorithms that can make best subset selection much faster [5]. A similar approach involves so-called “Akaike weights” [6,7]. With this method, importance is based on the sum of the Akaike weights for all the models that include a specific variable; the most important variables will have larger values of this sum. This approach is very slow because it requires that every possible model is fit. Thus, branch and bound algorithms that speed up the best subset selection process cannot be used. In fact, in the context of best subset selection, any algorithm for evaluating variable importance will likely become much slower as the number of variables increases, and if the algorithm requires fitting all candidate models, it will quickly become computationally prohibitive.

In this paper, we introduce a novel method for determining variable importance in the context of best subset selection that is relatively efficient computationally. Moreover, we believe that the proposed measure provides the most natural quantification of variable importance for best subset selection. An additional benefit of our measure is that valid *p*-values can be computed based on the parametric bootstrap method. Calculating the *p*-values, however, can be a slow process. We therefore discuss approaches that can improve computational efficiency.

The remainder of the manuscript is structured as follows. Section 2 presents the proposed variable importance measure along with some relevant theoretical results. Section 3 presents a simulation study to verify the validity of *p*-values based on the variable importance values. In Section 4, the proposed methods are demonstrated in a modeling application where the goal is to predict percentage body fat based on easily obtained anthropometric measurements. In Appendix A and Appendix B, we explore computational issues, and illustrate how the method can be applied in the context of other variable selection algorithms, including heuristic procedures such as forward selection and backward elimination.

## 2. Variable Importance for Best Subset Selection

### 2.1. Proposed Methodology

We will focus on variable selection that is based on the following penalized log-likelihood measure:$$Q\left(\beta |X,y,\lambda \right)\overset{def}{=}-2l\left(\beta |X,y\right)+{\lambda ||\beta ||}_{0}.$$
where l(β|X,y) denotes the log-likelihood for a model with regression parameters β, ${\left||\beta |\right|}_{0}$ is the number of non-zero values in β, and λ is the penalty term. The β that minimizes this penalized log-likelihood measure is the best subset selection estimator (i.e., the estimator resulting from the selected model from best subset selection). λ is commonly pre-specified according to the penalization of an information criterion. Some of the most popular information criteria are the Akaike information criterion (AIC, [8]) where λ=2, the Bayesian information criterion (BIC, [9]) where λ=logn, and the Hannan–Quinn information criterion (HQIC, [10]) where λ=2kloglogn (we select k=1 for our usage of HQIC). The form of Q(β|X,y,λ) represents that of a general likelihood-based information criterion.

We propose the following quantity to assess variable importance for the *i*th set of variables:$$VI\left(i,\lambda |X,y\right)=\min_{\beta :\beta_{S_{i}}=0}Q\left(\beta |X,y,\lambda \right)-\min_{\beta :\beta_{S_{i}}\neq 0}Q\left(\beta |X,y,\lambda \right),$$
where $S_{i}$ is a set that contains the indices of the variables that are included in the *i*th variable set. Note that in many applications, these variable sets will each only contain a single variable. However, assessing variable importance for a set of variables can be useful when there is a natural way to group together some of the variables. One important instance of this occurs with categorical variables where multiple indicator variables are used to encode the levels of the variable. For ease of exposition, in much of the subsequent presentation, we will often describe the measure as pertaining to one variable as opposed to a variable set.

The variable importance measure is very similar to the traditional likelihood ratio test statistic to compare two models, except that it uses the criterion Q(β|X,y,λ) instead of −2l(β|X,y), and compares Q(β|X,y,λ) for the optimal model that includes a variable (or variable set) of interest to the optimal model that excludes the same variable (or variable set). The measure is bounded below by $-\lambda |S_{i}|$ and is not bounded above. When the measure is negative, this means that the best subset selection estimator for the information criterion does not include the variable(s). Large negative values provide strong evidence that the variable(s) should not be included in the final model. When the quantity is positive, the best subset selection estimator includes the variable(s). Large positive values of this quantity provide strong evidence that the variable(s) should be included in the final model. When this quantity is zero, then the inclusion or exclusion of the variable(s) does not impact the optimal value of the penalized log-likelihood measure.

Based on simulation results, it appears that the following modification of the variable importance measure has a null sampling distribution that is sometimes well approximated by a chi-square distribution with degrees of freedom equal to $|S_{i}|$:$$mVI\left(i,\lambda |X,y\right)=VI\left(i,\lambda |X,y\right)+\lambda |S_{i}|.$$

Thus, this modified variable importance measure could be used as a test statistic to assess the contribution of a particular variable (or variable set). The chi-squared approximation, however, breaks down when the variables are highly correlated. Instead, we can use a simulation-based approach to approximate the null distribution for each test statistic. We suggest the use of the parametric bootstrap. However, when we are calculating the statistic for a given variable, we use the model that contains all of the variables except for the variable(s) of interest to generate the new response variable. This method preserves the correlation between the variables, which facilitates an accurate characterization of the null distribution. The main downside of this approach is that calculating the test statistic once can take some time, so repeatedly calculating the test statistic to approximate a null distribution can be very time consuming.

Calculating *p*-values based on the modified variable importance measure could be very useful since the traditional method of obtaining *p*-values after performing best subset selection is well known to be biased and only provides *p*-values for variables included in the final model. Additionally, using *p*-values based on the full model is not advisable when best subset selection is employed.

### 2.2. Theoretical Results

In this subsection, we will prove that the modified variable importance measures are non-negative.

First, we define the active set A as the set that contains the indices of the non-zero coefficients and $A^{∁}$ as the set that contains the indices of the zero coefficients. We also define the following constructs:$$\begin{array}{cc} -2l_{A}\left(\beta |X,y\right) & =\min_{\beta }-2l\left(\beta |X,y\right)\;s.t.\;\beta_{A^{∁}}=0\\ Q_{A}\left(\beta |X,y,\lambda \right) & =-2l_{A}\left(\beta |X,y\right)+\lambda \left|A\right|. \end{array}$$

**Lemma** **1.**
*For models $A_{1}$ and $A_{2}$, such that $A_{1}\subset A_{2}$, the following is true:*

(1)
$$Q_{A_{2}}\left(\beta |X,y,\lambda \right)-Q_{A_{1}}\left(\beta |X,y,\lambda \right)\leq \lambda (|A_{2}\left|-\right|A_{1}\left|\right)$$



**Proof.** Let $A_{1}$ define model one and $A_{2}$ define model two, where $A_{1}\subset A_{2}$. Hence, model one is a submodel of model two. Note that since model one is subsumed by model two, we know that $-2l_{A_{2}}\left(\beta |X,y\right)\leq -2l_{A_{1}}\left(\beta |X,y\right)$. We may assert the following:
$$\begin{array}{cc} Q_{A_{2}}\left(\beta |X,y,\lambda \right)-Q_{A_{1}}\left(\beta |X,y,\lambda \right) & =-2l_{A_{2}}\left(\beta |X,y\right)+\lambda |A_{2}|-(-2l_{A_{1}}\left(\beta |X,y\right)+\lambda |A_{1}\left|\right)\\ & \leq -2l_{A_{1}}\left(\beta |X,y\right)+\lambda |A_{2}|-(-2l_{A_{1}}\left(\beta |X,y\right)+\lambda |A_{1}\left|\right)\\ & =\lambda (|A_{2}|-|A_{1}|) \end{array}$$□

**Theorem** **1.**
*The modified variable importance measures are non-negative for X, y, λ, and i.*


**Proof.** Let $A_{1}$ be the active set for the optimal model that does not contain the *i*th variable set and let $A_{1i}$ be the same active set, but with the *i*th variable set included. Also, let $A_{2}$ be the active set for the optimal model that contains the *i*th variable set. $A_{1}\subset A_{1i}$, so we have the following:
$$\begin{array}{cccc} Q_{A_{1i}}\left(\beta |X,y,\lambda \right)-Q_{A_{1}}\left(\beta |X,y,\lambda \right) & \leq \lambda (|A_{1i}|-|A_{1}|) & & \mathrm{by}\;(1)\\ ⟹Q_{A_{1i}}\left(\beta |X,y,\lambda \right)-Q_{A_{1}}\left(\beta |X,y,\lambda \right) & \leq \lambda |S_{i}|\\ ⟹Q_{A_{1}}\left(\beta |X,y,\lambda \right)-Q_{A_{1i}}\left(\beta |X,y,\lambda \right) & \geq -\lambda |S_{i}| \end{array}$$We also have $Q_{A_{2}}\left(\beta |X,y,\lambda \right)\leq Q_{A_{1i}}\left(\beta |X,y,\lambda \right)$, since $A_{2}$ is the active set of the model with the minimum penalized log-likelihood measure that includes the *i*th variable set and $A_{1i}$ also has the *i*th variable set. Hence, we can establish that
$$\begin{array}{cc} Q_{A_{1}}\left(\beta |X,y,\lambda \right)-Q_{A_{2}}\left(\beta |X,y,\lambda \right) & \geq Q_{A_{1}}\left(\beta |X,y,\lambda \right)-Q_{A_{1i}}\left(\beta |X,y,\lambda \right)\\ & \geq -\lambda |S_{i}|\\ ⟹Q_{A_{1}}\left(\beta |X,y,\lambda \right)-Q_{A_{2}}\left(\beta |X,y,\lambda \right)+\lambda \left|S_{i}\right| & \geq 0 \end{array}$$The result then follows from the definition of the modified variable importance measure. □

## 3. *p*-Value Simulations

In this section, we provide a set of simulation results that validate the parametric bootstrap approach to calculating *p*-values based on the modified variable importance measure. These results also invalidate the omnibus use of the chi-square approximation, which is especially problematic when there is strong correlation between the variables and when the penalty term grows with the sample size.

The setup for the simulations used in this section is as follows. First, we generate various linear regression models by initially defining a regression parameter vector β for six covariates as $\beta^{⊤}=\left(\begin{array}{cccccc} 2 & 3 & 4 & 5 & 6 & 7 \end{array}\right)$. To create null effects for each of our models, we randomly set three of these six regression coefficients to zero. With this approach, the relative importance of the three retained non-null effects will vary from one model to the next.

For the generation of the covariate vectors $x_{i}$, we define $\Sigma =\rho J_{6}+\left(1-\rho \right)I_{6}$, where $J_{6}$ is a 6 by 6 matrix of all ones. We then generate the vectors as $x_{i}∼N_{6}\left(1,\Sigma \right).$ With $\mu_{i}=x_{i}^{⊤}\beta +1$, we generate the outcome variables as $y_{i}∼N\left(\mu_{i},\sigma^{2}\right)$. We determine the variance $\sigma^{2}$ by considering the relation $\sigma^{2}=\beta^{⊤}\Sigma \beta /\mathrm{SNR}$, where SNR denotes the signal-to-noise ratio, commonly defined as $Var\left(x^{⊤}\beta \right)/\sigma^{2}$ for linear regression.

To set the variance $\sigma^{2}$, we use a SNR of about 0.4286, which corresponds to a coefficient of determination of $R^{2}=0.3$. We considered multiple different values of SNR, but we decided on this value of SNR because it proved to be the most problematic for our methods. This SNR would therefore illustrate the efficacy of our method in a “worst-case” scenario.

### 3.1. Simulated Distributions

For the following simulation sets, we generate 10,000 linear regression models as described above, with ρ∈{0,0.9}. For each of these models, we generate 1000 observations. We then calculate the modified variable importance measures for each variable that has a zero coefficient. Next, we collect all 30,000 modified variable importance values corresponding to these null effects and plot their empirical distribution relative to a $\chi_{1}^{2}$ distribution. We repeat this process for AIC, BIC, and HQIC.

In the set of simulations shown in Figure 1, it appears that the modified variable importance measures corresponding to each of the information criteria are roughly chi-squared distributed under independence. However, it appears that the modified variable importance measures are not approximately $\chi_{1}^{2}$ distributed when there is correlation between the variables. The distribution appears to deviate further from a $\chi_{1}^{2}$ as λ increases when there is correlation.

### 3.2. Type 1 Error Rates

For the following sets of results, we again generate 10,000 linear regression models as previously described, with ρ∈{0,0.9} and a sample size of 1000 observations for each model. For each of these simulated regressions, we calculate the test statistics and *p*-values for each variable. We then use the *p*-values from the coefficients that were truly zero to calculate the type 1 error rates for each of the different scenarios.

#### 3.2.1. Naive Approach

Here, we show the type 1 error rates that would result from treating the modified variable importance measures as if they were $\chi_{1}^{2}$ distributed. We consider rates based on the *p*-values corresponding to AIC, BIC, and HQIC.

From the simulation results in Figure 2, it appears that the distribution of the test statistics is well approximated by a chi-square distribution under independence. In large-sample settings, note that all of the tests appear to have the correct type 1 error rate for the independent variables results. However, we can see that the type 1 error rates for the tests with BIC and HQIC are much too large when the variables are highly correlated with ρ=0.9. The type 1 error rate with AIC does appear to be much closer to the desired level, but it is still slightly too large.

From Figure 3, we can see that the type 1 error rates from using this approach with BIC and HQIC tend to get larger as the correlation between variables increases. It appears that the type 1 error rates when using this method with AIC are slightly larger than desired, but are still fairly close to the desired error rates.

#### 3.2.2. Bootstrapping Approach

We propose the use of a parametric approach that can be employed to approximate the null distribution of the modified variable importance measure. For a given variable, we generate a new realization of the response variable based on the model that includes all variables except for the current variable of interest. We then calculate the modified variable importance measure for the current variable of interest. We carry out this calculation for each variable, repeating the process many times to empirically approximate null distributions for the importance measures.

Here, we show the type 1 error rates that would result from using the parametric bootstrap approach described above. The null distribution of the modified variable importance measure for each variable is estimated with 100 bootstrapped replicates in each simulation.

From the simulation results in Figure 4 and Figure 5, we see that the parametric bootstrap approach gives approximately correct type 1 error rates even with BIC and HQIC under large correlation. We do see some slight deviations from the desired type 1 error rates under atypically high correlation. However, even under a very extreme correlation of 0.99 between all of the variables, the type 1 error rates are still within about 0.03 of the desired rates.

It appears that the parametric bootstrap approach can be used to obtain valid *p*-values from the modified variable importance measures. However, this method is much more computationally expensive than using a $\chi_{1}^{2}$ distribution because we need to perform best subset selection 2*p* ∗ nboot times, where nboot denotes the number of bootstrap replicates and *p* denotes the number of variables. Thus, in Appendix A, we discuss approaches for improving computational efficiency.

## 4. Body Fat Dataset Example

In this section, we demonstrate the utility of the proposed methods in an application. The goal is to build a regression model to predict the body fat percentage for men based on the following variables: age (years), weight (pounds), height (inches), and ankle (cm), bicep (cm), chest (cm), forearm (cm), hip (cm), knee (cm), neck (cm), thigh (cm), waist (inches), and wrist (cm) circumference. Our dataset can be found at https://dasl.datadescription.com/datafile/bodyfat/ (accessed on 2 August 2024) and comprises 250 observations.

In the context of linear regression, we will consider the use of best subset selection with AIC, BIC, and HQIC. We will demonstrate the utility of the proposed variable importance measure, as well as *p*-values based on the modified version of this measure, calculated using the parametric bootstrap. We will calculate both the variable importance measures and the *p*-values for AIC, BIC, and HQIC, and we will compare these results to what one would obtain using a conventional approach based on using the full model.

### 4.1. Coefficient Estimates

The standardized coefficient estimates resulting from the full model and from best subset selection using each of the information criteria are featured in Table 1. From these results, we can see that waist and wrist circumference appear to be the most important variables because they have large coefficient estimates (in magnitude) and they are both included in all of the models.

We can also see that weight is included in the model selected by BIC, but is not included in the model selected by AIC or HQIC. This is a very interesting result, because we would expect the variables that are included in the model selected by BIC to be included in the models favored by AIC and HQIC, since BIC has the largest penalty term and therefore favors more parsimonious models. We suspect that this inconsistency occurs because weight is highly correlated with many of the body measurement variables, yet does not seem to explain much additional variability in the outcome when these other variables are included in the model. Thus, when the majority of the body measurement variables are excluded from the model, weight can be a very useful variable to include. However, when a number of these body measurement variables are included, weight does not seem to provide much additional benefit. This can also be seen because the coefficient for weight is more than 10 times larger (in magnitude) in the BIC model as compared to the full model.

### 4.2. Variable Importance

Variable importance assessments resulting from each method are displayed in Table 2. The variable importance measures for the full model are taken to be the squared Wald test statistics for each of the variables. These measures reflect some of the same conclusions that we noted based on the coefficient estimates. Namely, for each method, waist circumference is the most important variable by far, and wrist circumference is the second most important variable. These results also show that weight is very important for best subset selection with BIC, and is somewhat important for best subset selection with HQIC, but is much less important for the other methods.

### 4.3. p-Values

The *p*-values resulting from each method, along with Wald *p*-values based on the selected models, are featured in Table 3. Note that the Wald *p*-values based on the selected models differ, often substantially, from the Wald *p*-values arising from the full model and from the *p*-values based on variable importance. The difference in these *p*-values becomes more pronounced as the penalty term of the selection criterion increases. For instance, in the model selected by BIC, weight has a Wald *p*-value of 1.3 × 10^−4^. However, the *p*-value for weight based on the full model is 0.895 and the *p*-value for weight based on variable importance with BIC is 0.224.

An interesting result from these analyses is that weight is the most important variable for best subset selection based on BIC, but it has a relatively large associated *p*-value based on simulation. This is likely a consequence of the high correlation between weight and the other body measurements. The *p*-value suggests that weight does not have a strong effect in the presence of these other measurements. Weight is correlated with body measurements that each explain some variation in the response, and therefore has a small coefficient estimate when several of these variables are included in the model, yet has a large coefficient estimate when the majority of these variables are not included in the model.

## 5. Discussion

In this paper, we develop and investigate a novel method for assessing variable importance when performing best subset selection. Our measure is specifically designed for the context of best subset selection, and provides a cogent approach for quantifying variable importance based on comparing two values of an information criterion: one for the optimal model that includes a variable (or variable set) of interest, and the other for the optimal model that excludes the same variable (or variable set). Our modified variable importance measure may be used in conjunction with the parametric bootstrap to calculate associated *p*-values, providing an inferential approach that is defensible when best subset selection is performed. The methods presented in this paper are available from the BranchGLM (version 3.0.0) R (version 4.4.1) [11] package, which is available on CRAN.

Because our variable importance measure was developed expressly for the purpose of best subset selection, we note that it may yield results that are different from those based on bootstrap selection frequencies or Akaike weights, each of which can be employed in the context of model averaging. We also note that data splitting provides an alternative to our approach for calculating *p*-values that are valid under best subset selection. Again, however, data splitting may yield different *p*-values compared to our method. Although both *p*-values are used to detect null and non-null effects, the *p*-values associated with our method are based on the importance of an effect across all possible models, whereas the conventional *p*-value only considers a specific model.

## Figures and Tables

**Figure 1 entropy-26-00801-f001:**
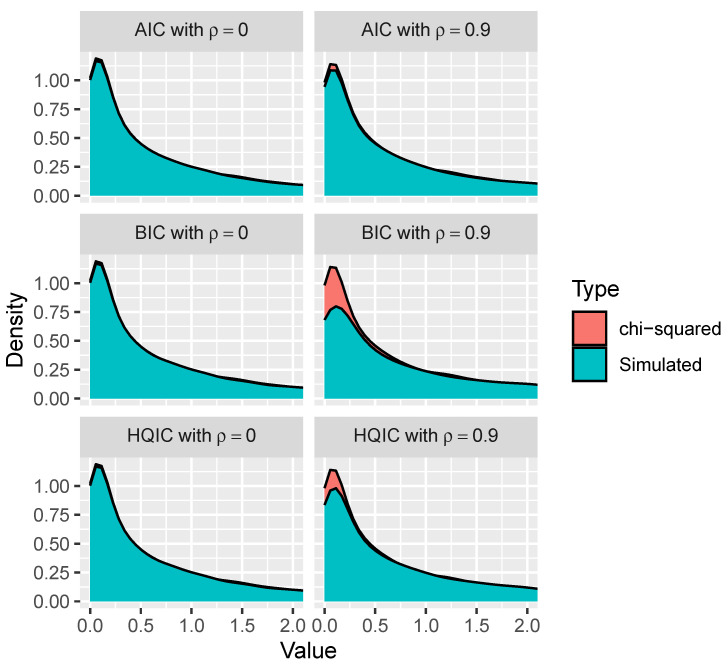
Test statistic simulations with n=1000 and ρ∈{0,0.9}.

**Figure 2 entropy-26-00801-f002:**
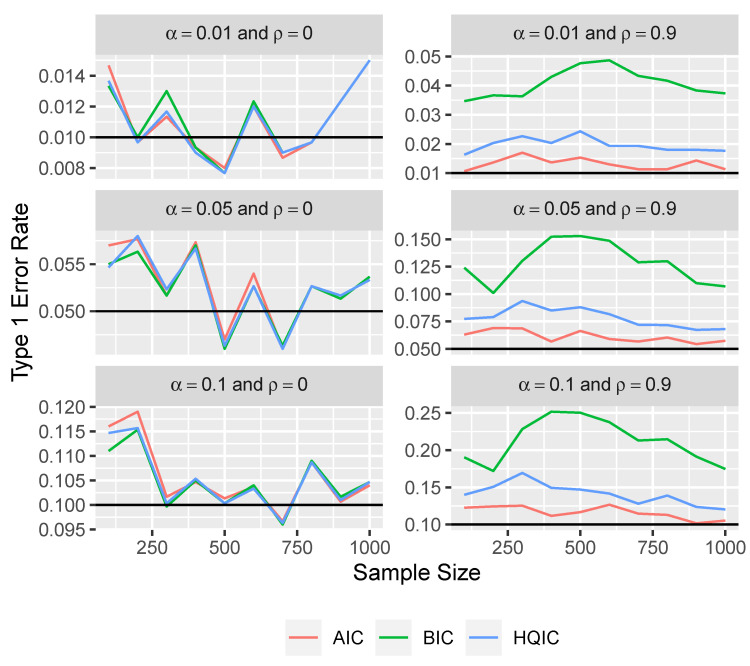
Type 1 error rate versus sample size for three levels of significance and for ρ∈{0,0.9} using the naive approach.

**Figure 3 entropy-26-00801-f003:**
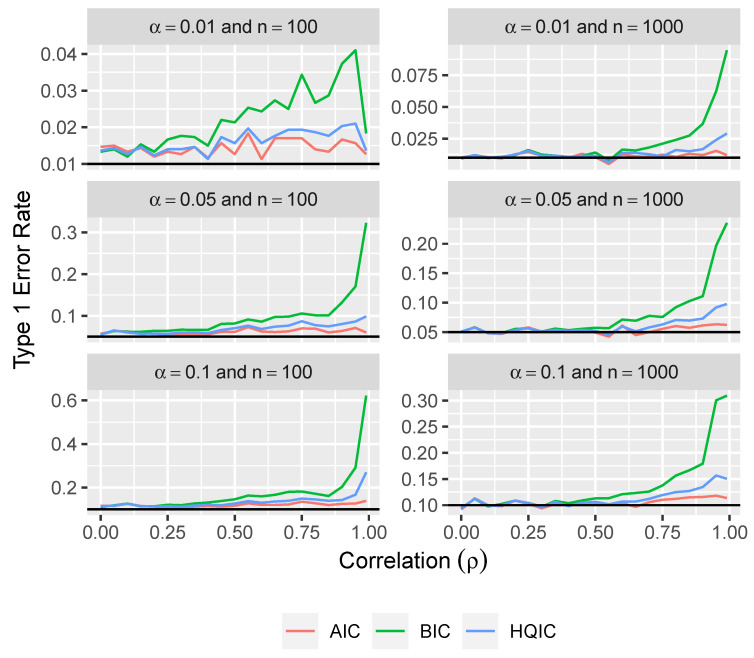
Type 1 error rate versus correlation for three levels of significance and for two sample sizes using the naive approach.

**Figure 4 entropy-26-00801-f004:**
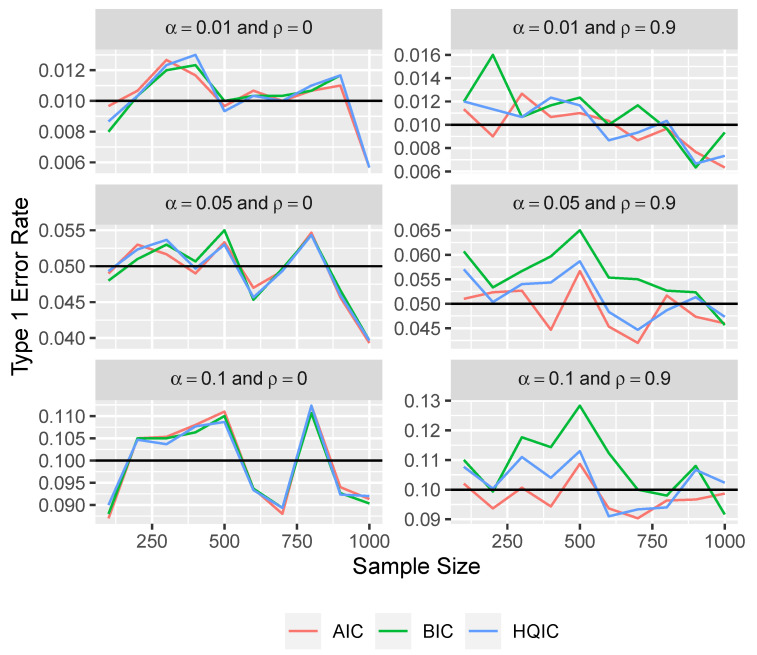
Type 1 error rate versus sample size for three levels of significance and for ρ∈{0,0.9} using the parametric bootstrap approach.

**Figure 5 entropy-26-00801-f005:**
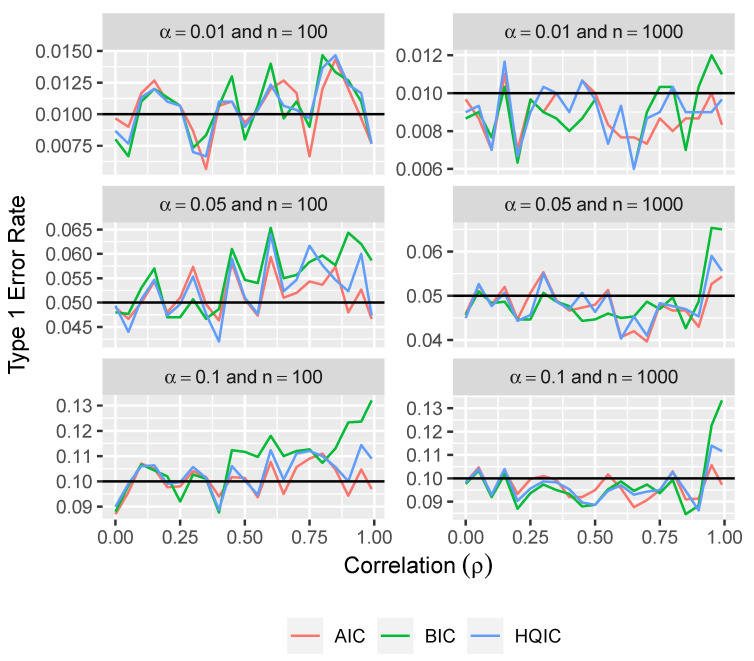
Type 1 error rate versus correlation for three levels of significance and for two sample sizes using the parametric bootstrap approach.

**Table 1 entropy-26-00801-t001:** Standardized coefficient estimates from each method.

	Full	AIC	HQIC	BIC
(Intercept)	19.108	19.032	19.032	19.032
Age	0.898	0.771	0.709	0.000
Weight	−0.233	0.000	0.000	−2.535
Height	−0.701	−0.841	−0.845	0.000
Neck	−0.874	−0.740	0.000	0.000
Chest	−1.108	−1.083	0.000	0.000
Waist	9.152	8.886	7.870	9.802
Hip	−1.038	0.000	0.000	0.000
Thigh	0.741	0.000	0.000	0.000
Knee	−0.106	0.000	0.000	0.000
Ankle	0.285	0.000	0.000	0.000
Bicep	0.555	0.000	0.000	0.000
Forearm	0.525	0.729	0.000	0.000
Wrist	−1.654	−1.571	−1.747	−1.254

**Table 2 entropy-26-00801-t002:** Variable importance from each method.

	Full	AIC	HQIC	BIC
Age	5.144	3.918	4.230	4.573
Weight	0.017	1.172	2.604	6.470
Height	2.063	2.828	4.230	4.573
Neck	2.827	2.155	1.954	1.827
Chest	1.679	2.355	2.446	2.373
Waist	103.820	84.424	85.880	88.933
Hip	1.316	1.216	0.590	1.746
Thigh	1.129	1.216	1.387	1.468
Knee	0.038	0.049	0.155	0.296
Ankle	0.659	0.762	0.772	0.834
Bicep	1.324	1.980	2.604	3.760
Forearm	1.672	2.020	1.524	2.680
Wrist	12.348	9.855	10.167	9.882

**Table 3 entropy-26-00801-t003:** *p*-values from each method and Wald *p*-values from the selected models.

	Full	AIC	AIC (Wald)	HQIC	HQIC (Wald)	BIC	BIC (Wald)
Age	0.024	0.058	0.015	0.064	0.019	0.077	
Weight	0.895	0.452		0.387		0.224	<0.001
Height	0.152	0.151	0.008	0.107	0.008	0.143	
Neck	0.094	0.164	0.157	0.207		0.205	
Chest	0.196	0.151	0.109	0.159		0.184	
Waist	<0.001	<0.001	<0.001	<0.001	<0.001	<0.001	<0.001
Hip	0.252	0.338		0.535		0.271	
Thigh	0.289	0.324		0.327		0.308	
Knee	0.847	0.845		0.746		0.641	
Ankle	0.418	0.409		0.377		0.392	
Bicep	0.251	0.187		0.126		0.103	
Forearm	0.197	0.187	0.070	0.237		0.120	
Wrist	<0.001	0.003	<0.001	0.001	<0.001	0.003	0.002

## Data Availability

The body fat dataset used for the application in this paper is available at https://dasl.datadescription.com/datafile/bodyfat/ (accessed on 2 August 2024).

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
