# Peer review of "Assessing Variable Importance for Best Subset Selection"

_entropy, 2024, doi:10.3390/e26090801_

Round 1
Reviewer 1 Report
Comments and Suggestions for Authors
See the attached pdf file.

Reviewer 2 Report
Comments and Suggestions for Authors
Review of
J. Seedorff and J. Cavanaugh Assessing Variable Importance for Bet Subset Selection.
Some details:
Line 28 “it is well known” — give an appropriate reference
Lines 120-121 (A) and (A∁) — A and A∁, as used in the mathematical expressions below.
Lines 132-134. Explain, it is not clear.
Lines 140-141 What is the rationale for that choice?
ρ ∈ {0, 0.9} is enough to reach conclusions?
Lines 172-173 “as the correlation between variables increases.” — there are only results for ρ ∈ {0, 0.9}, so the claim is excessive.
Line 191 — what is p here? The fact that in the Appendix p is used for the number of models or of variables, but so far the nonitalic symbol p had only been used in p-values.
Line 280 … p variables — I think that it would be better to use another symbol for the number of variables, the use of p-values and p variables and p models is in my opinion inappropriate.
In the Figures, why rho and alpha instead of r and a?
General comments: The paper is well written, with judicious comments. From the fact that the presentation of results is restricted to ρ ∈ {0, 0.9}, some issues seem inconclusive. Figures should be commented with more details, since the deviation from the declared value of a is large, namely for large sample sizes.
Its subject seems to be very loosely related to the aims and scopes of Entropy, and so I hesitate to recommend its acceptation.
